# Recent Advances in the Development and Clinical Use of HER2 Inhibitors in Non-Small Cell Lung Cancer

**DOI:** 10.3390/biom15101443

**Published:** 2025-10-12

**Authors:** Richy Ekyalongo, Toshimitsu Yamaoka, Junji Tsurutani

**Affiliations:** Advanced Cancer Translational Research Institute, Showa Medical University; 1-5-8 Hatanodai, Shinagawa-ku, Tokyo 142-8555, Japan; richyekyalongo@gmail.com (R.E.); tsurutaj@med.showa-u.ac.jp (J.T.)

**Keywords:** NSCLC, breast cancer, anti-HER2 inhibitors, HER2 mutations, HER2 amplification

## Abstract

Alterations in the human epidermal growth factor receptor 2 (*HER2*) gene are well-recognized oncogenic drivers and therapeutic targets in non-small cell lung cancer (NSCLC). The first anti-HER2 inhibitor, trastuzumab-deruxtecan, was approved for previously treated advanced NSCLC with *HER2* mutations, which accounts for 2–4% of NSCLC. The first anti-HER2 antibody, trastuzumab, was approved for HER2-positive metastatic breast cancer in 1998, and a combination therapy comprising trastuzumab, pertuzumab, and docetaxel demonstrated efficacy in the first-line setting. Some EGFR-tyrosine kinase inhibitors (TKIs) have been evaluated as pan-HER TKIs but have shown limited benefits in *HER2*-altered NSCLC. However, HER2-specific TKIs, such as zongertinib and BAY2927088, have demonstrated encouraging results. Zongertinib was the first HER2-specific TKI to be approved by the FDA in 2025 for previously treated *ERBB2*-mutated advanced NSCLC. In this narrative review, we have summarized the latest research on the biology of HER2 signaling, HER2 alterations, HER2-targeting therapies, and challenges of treating HER2-overexpressing or -mutated NSCLC. Despite different targets of *HER2* mutations in NSCLC and HER2 amplification/overexpression in breast cancer, the development of HER2-targeting agents has been more advanced in breast cancer than in NSCLC. Therefore, pivotal clinical studies in breast cancer may help in identifying more effective therapies for NSCLC.

## 1. Introduction

Non-small cell lung cancer (NSCLC) is the leading cause of death among all cancers. However, the introduction of molecular targeted therapies for NSCLC has significantly improved survival [1]. Mutations, such as point mutations, deletions, insertions, or translocations in driver oncogenes, promote tumor growth and invasiveness, resulting in poor prognosis [2]. In 2004, the epidermal growth factor receptor-tyrosine kinase inhibitor (EGFR-TKI) gefitinib was approved for the treatment of patients with NSCLC with *EGFR* activating mutations [3]. Twenty years later, in 2024, trastuzumab-deruxtecan (T-DXd), an antibody-drug conjugate (ADC) targeting the human epidermal growth factor receptor 2 (HER2), which belongs to the EGFR family, was approved for patients with NSCLC harboring *HER2* mutations [4]. Alterations in the *HER2* gene have been recognized as an oncogenic driver in many cancers, including breast, gastric, colon, and lung cancers [5]. In 1998, trastuzumab, an anti-HER2 antibody, was approved for HER2-positive breast cancer, which accounts for approximately 20% of all breast cancer cases [6]. It is challenging to develop therapeutic strategies for HER2-positive or HER2-amplified NSCLC because of the difficulty in identifying tumors whose growth mainly depends on HER2 signaling. In this narrative review, we carefully selected the most up-to-date and published information and comprehensively discussed the alterations, gene amplification, protein overexpression, and mutations of *HER2* in NSCLC. Additionally, we have highlighted the difficult trajectory of developing HER2 inhibitors, such as TKIs, antibodies, and ADCs, for patients with HER2-overexpressing or *HER2*-mutated NSCLC and compared it with the development of HER2-targeting treatment strategies for patients with metastatic breast cancer. Despite the different targets of *HER2* mutations in NSCLC and *HER2* amplification in breast cancer, the development of HER2-targeting agents and therapeutic strategies has been more advanced in breast cancer than in NSCLC. Therefore, it would be meaningful to understand the pivotal clinical studies in breast cancer to identify more effective therapies for NSCLC.

## 2. Biology of HER2 Signal Transduction

HER2 is a member of the EGFR family of tyrosine kinase receptors, which includes HER1 (EGFR), HER2, HER3, and HER4 [7]. HER2 plays a critical role in cell proliferation, survival, and differentiation [8]. Unlike the other members of the EGFR family, HER2 has no known direct ligands. Instead, it functions primarily through dimerization with other HER receptors to form potent signaling complexes [9]. HER2-containing heterodimers, particularly HER2-HER3, generate strong downstream signaling via the PI3K/AKT and MAPK pathways, contributing to cell growth and survival [8]. Moreover, HER2 dimerization is not limited to HER2-HER3 complexes. HER2 can form homodimers or heterodimers with other EGFR family members, thereby influencing the activation of downstream pathways [10]. HER3, which lacks or has weak intrinsic kinase activity, relies heavily on HER2 for signal transduction [11]. Because HER3 contains p85-binding motifs, HER3 can directly bind to and activate p85 of PI3K. HER2/HER3 hetero-dimerization is particularly important for activating the PI3K/AKT pathway, which supports cell survival and therapeutic resistance (Figure 1) [12].

The gene encoding HER2 (also known as ERBB2) is located on chromosome 17q21 and produces a 185 kDa transmembrane glycoprotein [13]. Low levels of HER2 are typically present in normal cells; however, its overexpression, amplification, or mutation can lead to uncontrolled cell growth and tumor formation. The HER2 receptor consists of three key domains: an extracellular ligand-binding domain, single transmembrane domain, and intracellular tyrosine kinase domain [14]. Upon dimerization, the tyrosine kinase domain is autophosphorylated, activating downstream pathways that regulate cell cycle progression, angiogenesis, and inhibition of apoptosis. The dysregulation of this signaling is a hallmark of HER2-driven cancer [15]. In NSCLC, the *ERBB2* mutation, which is the therapeutic target of approved inhibitors, can activate the HER2 kinase, resulting in constitutive activation of downstream signaling. In breast cancer, HER2 overexpression, which is also the therapeutic target of approved inhibitors, can be activated by HER2 homodimerization and heterodimerization with other EGFR family members, and lead to the autophosphorylation of these tyrosine residues in intracellular domain to initiate a signal cascade.

## 3. HER2 Alterations and Prognosis in Patients with NSCLC and Breast Cancer

HER2 plays a pivotal role in the pathogenesis of various cancers, notably NSCLC, breast cancer, gastric cancer, colon cancer, gynecologic cancer, and others [5]. Alterations in HER2 include gene mutation, gene amplification, and protein overexpression. *ERBB2* gene amplification and mutation are generally mutually exclusive, suggesting that they are independent oncogenic drivers of tumorigenesis, and these alterations may be responsible for the different oncogenesis, biological properties, and clinical features of various cancers [5,16].

### 3.1. ERBB2 Mutations

In NSCLC, *ERBB2* mutations are identified in approximately 2–4% of cases and located predominantly in the αC-β4 loop of the kinase domain in the cytoplasmic region (Figure 2) [17,18]. Most *ERBB2* mutations in NSCLC are in-frame insertions in exon 20 (ex20ins). The most common mutation is Y772_A775dup, which is a duplication/insertion. Y772_A775dup represents three ex20ins: E770_A771insAYVM, A771_Y772insYVMA, and A775/G776insYVMA (Figure 3). G778_P780dup represents two insertions: V777/G778insGSP and P780_Y781insGSP. HER2 ex20ins include Y772_A775dup, G778_P780dup, and G776delinsVC, and are observed in nearly 90% of *ERBB2*-mutated lung cancers [19]. Whole-genome sequencing (WGS) data from all non-overlapping studies were exported to NSCLC cBioPortal (available at http://www.cbioportal.org (accessed on 21 April 2025)), and it was determined that *ERBB2* mutations occur in 3.5% of NSCLC cases (Figure 2). All patients harboring Y772_A775dup and G778_P780dup were diagnosed with lung adenocarcinoma and were predominantly females, nonsmokers, or light smokers, similar to patients with lung adenocarcinoma with *EGFR* mutations [20]. These mutations are oncogenic drivers that promote malignant transformation and tumor progression [21]. Notably, *ERBB2* mutations in NSCLC are typically mutually exclusive of other driver mutations, such as *EGFR*, *KRAS*, and *ALK* rearrangements, highlighting their potential to serve as distinct therapeutic targets [19]. Patients with advanced lung cancer with the *ERBB2* mutation have previously been reported to have a poor prognosis [18,22].

*ERBB2*-activating mutations that function as oncogenic drivers have been reported in approximately 4% of breast cancers, [23]. *ERBB2* mutations commonly occur in the absence of *ERBB2* amplification and are usually identified in invasive lobular breast cancer [24,25], which accounts for 15% of the estrogen receptor positive (ER+) subtype [26]. WGS data from all nonoverlapping studies were selected and exported to Breast Cancer (http://www.cbioportal.org (accessed on 21 April 2025)). *ERBB2* mutations were identified in 3.4% of breast cancer cases. Most *ERBB2* mutations in breast cancer were found in the tyrosine kinase domain of exons 19 and 20. Moreover, point mutations were the most frequent, followed by in-frame insertions/deletions. In the tyrosine kinase domain of *HER2*, L755S and V771L accounted for nearly 50% of all mutations. Exon 20 insertions (including G778_P780dup and Y772_A775dup) in *ERBB2* accounted for nearly 15% of all mutations in patients with breast cancer (Figure 3). Previous studies have reported that *ERBB2* mutations are associated with a poor prognosis [27,28]. Therefore, the primary focus of breast cancer research has been on *ERBB2* amplification and *HER2* overexpression rather than mutations, and there are no standard therapies targeting *ERBB2* mutations in breast cancer. Although *ERBB2* mutations have been recognized as potential therapeutic targets, they are not standard targets in breast cancer yet. Croessman et al. found that the HER2 L755S mutation promotes HER2-HER3 heterodimerization, leading to PI3K/AKT/mTOR axis hyperactivation and estrogen resistance in ER+ breast cancer, indicating that dual blockade of the HER2 and ER pathways is required for the treatment of *ER+*/*HER2*-mutated breast cancer [29]. A previous study has also detailed the impact of *ERBB2* specific mutations and activity in breast cancer [30].

### 3.2. ERBB2 Amplification and HER2 Protein Overexpression

*ERBB2* amplification leads to an increased number of *ERBB2* copies, frequently resulting in HER2 protein overexpression [31]. In NSCLC, *ERBB2* amplification occurs in approximately 2–5% of cases. *ERBB2* amplification was reportedly more common in men and smokers in a Japanese cohort study [32]. Although *ERBB2* amplification/HER2 overexpression is a clear target for anti-HER2 therapy in advanced breast and gastroesophageal cancers, there is no officially approved therapy for patients with advanced NSCLC with *ERBB2* amplification/HER2 overexpression. Currently, there is no universally accepted standard definition of *ERBB2* amplification or HER2 overexpression in NSCLC. Generally, a target gene/chromosome ratio greater than two, based on fluorescence in situ hybridization (FISH), has been recognized to represent gene amplification [33,34]. Establishing the definition of *ERBB2* amplification in patients with NSCLC who may benefit from anti-HER2 therapy is a critical issue. Next-generation sequencing (NGS) is now routinely employed to detect driver mutations in patients with NSCLC. Therefore, it is reasonable to employ NGS or FISH for detecting *ERBB2* amplification and immunohistochemistry (IHC) for HER2 overexpression [33]. Odintsov et al. analyzed next-generation DNA sequencing data (OncoPanel) from over 5000 patients and defined high-level *ERBB2* amplification as greater than six copies. High-level *ERBB2* amplification was identified in 0.9% of lung adenocarcinomas and correlated with HER2 protein overexpression. Interestingly, 50% of patients with NSCLC with *ERBB2* amplification did not have other mitogenic drivers, 25% of the patients with *ERBB2*-amplified NSCLC carried *ERBB2* mutations, and the other 25% had receptor tyrosine kinase (RTK) and MAPK pathway component mutations (including patients who previously received targeted therapies such as EGFR-TKI) [35]. Although *ERBB2* amplification should be considered a distinct molecular target, its underlying mechanisms are diverse. From a therapeutic perspective, NSCLC with *ERBB2* amplification should be defined as a specific subgroup.

In contrast, *ERBB2* amplification/HER2 overexpression is observed in approximately 20–30% of primary breast tumors. Amplification/overexpression serves as a critical predictor of aggressive disease and reduced survival [36]. HER2 status is important for selecting the optimal therapy for breast cancer. HER2 overexpression is assessed by IHC using the monoclonal antibody CB11, and *ERBB2* amplification is determined by FISH. IHC reactions of 3+ are considered HER-2 positive, while IHC reactions of 0 and 1+ are considered HER-2 negative. According to the American Society of Clinical Oncology and the College of American Pathologists (ASCO/CAP) guidelines, HER2-positive tumors are defined as either IHC (3+), indicating HER2 protein overexpression, or FISH (+), indicating *ERBB2* amplification [37].

## 4. Clinical Utility of HER2 Tyrosine Kinase Inhibitors

### 4.1. Afatinib and Dacomitinib as Pan-HER Inhibitors

The therapeutic potency of the HER2-TKIs afatinib and dacomitinib is summarized in Table 1. These HER2-TKIs have been tested in pretreated patients with *ERBB2*-mutated NSCLC. Afatinib and dacomitinib are irreversible pan-HER inhibitors that block the activation of both homodimers and heterodimers within the HER family [38]. Afatinib was approved for *EGFR*-mutated NSCLC in 2013 by the FDA [39] and dacomitinib was approved for patients with NSCLC with common *EGFR* mutations, including *EGFR* exon 19 deletion or exon 21 L858R substitution [40]. Therefore, afatinib and dacomitinib have been evaluated for patients with NSCLC harboring *ERBB2* mutations. However, all prospective clinical trials of afatinib and dacomitinib have demonstrated limited efficacy with an objective response rate (ORR) of 0–11.5% and median progression-free survival (mPFS) of 2.8–4.3 months [41,42,43,44] (Table 1). A retrospective study showed that chemotherapy could bring more benefit to patients with *ERBB2*-mutated NSCLC than afatinib, especially those with the most common type of HER2 exon 20 insertion of A775_G776insYVMA [45]. A775_G776insYVMA in HER2 exon 20 can activate ligand-independent kinase signaling by changing the conformational landscape of HER2 kinase, resulting in reduced potential inhibition by afatinib and dacomitinib [20].

### 4.2. Neratinib, Pyrotinib, and Tucatinib for Patients with ERBB2-Mutated NSCLC and HER2-Positive Breast Cancer

Currently, there is only one TKI that targets *ERBB2* mutations, zongertinib, and no TKI that targets *ERBB2* amplification in patients with NSCLC. In contrast, for the treatment of HER2-positive metastatic breast cancer (MBC), several HER2-TKIs, such as neratinib, lapatinib, tucatinib, and pyrotinib, have been approved [46] (Table 2).

Neratinib is an irreversible, non-selective pan-HER family TKI. The activity of neratinib in *ERBB2*-mutated NSCLC was investigated in two phase II trials: SUMMIT and PUMA-NER-4201 (Table 3). In the SUMMIT study, which was a basket trial, 26 patients with *ERBB2*-mutated NSCLC were enrolled and treated with neratinib, resulting in an ORR of 3.8% and mPFS of 5.5 months, demonstrating minimum effectiveness [47]. Notably, a dual inhibition therapeutic strategy comprising neratinib + temsirolimus, which is an mTOR inhibitor, yielded a moderate ORR of 19% and PFS of 4.1 months, slightly higher than an ORR of 0% and PFS of 3.0 months with single arm neratinib [48]. In the NALA study on HER2-positive-MBC, neratinib plus capecitabine yielded an ORR of 32.8% and PFS of 8.8 months, whereas lapatinib with capecitabine resulted in an ORR of 26.7% and PFS of 6.6 months [49,50]. Based on this phase III study, the FDA approved neratinib in combination with capecitabine for adult patients with advanced or metastatic HER2-positive breast cancer who had received two or more prior anti-HER2-based regimens in the metastatic setting. Thus, neratinib is effective in HER2 amplified MBC, but not in *ERBB2*-mutated NSCLC. This highlights the requirement for selective HER2 inhibitors in *ERBB2*-mutated NSCLC.

Pyrotinib is an irreversible EGFR/HER2/HER4 small molecule inhibitor that has also shown efficacy in *ERBB2* mutated NSCLC with an ORR of 19.2–30.0% and median PFS of 5.6–6.9 months, with a manageable toxicity profile in different studies as a single agent [51,52]. In HER2 positive-MBC, pyrotinib plus capecitabine in the PHOEBE study yielded an ORR of 67.2% and PFS of 12.5 months, whereas lapatinib with capecitabine resulted in an ORR of 51.5% and PFS of 6.8 months [53]. In combination with pyrotinib, apatinib, an oral small-molecule TKI that selectively targets VEGFR2 has shown benefits in patients with advanced NSCLC when combined with the EGFR-TKI gefitinib and has been approved in China [54]. The combination of pyrotinib and apatinib showed encouraging efficacy with an ORR of 51.5% and mPFS of 6.9 months [55]. Dual HER2-targeted regimens, such as tucatinib in combination with trastuzumab and capecitabine, have demonstrated efficacy in HER2-positive breast cancer [56]. Inetetamab is a humanized, recombinant anti-HER2 monoclonal antibody that is a biosimilar product of trastuzumab [57]. The combination of pyrotinib and inetetamab was examined for treating patients with *ERBB2*-mutated NSCLC in a phase Ib trial and showed an ORR of 25% and mPFS of 5.5 months, along with a manageable safety profile and promising anti-tumor activity [58].

Tucatinib is a selective HER2-TKI. In the HER2 CLIMB study on HER2-positive breast cancer, when tucatinib was combined with trastuzumab and capecitabine, the PFS and overall survival (OS) were significantly prolonged compared to those in the placebo + trastuzumab and capecitabine group. Furthermore, tucatinib exhibited favorable penetration in the brain [56]. In 2020, the FDA approved a combination regimen of tucatinib, trastuzumab, and capecitabine for locally advanced unresectable or metastatic HER2-positive breast cancer. However, although tucatinib has been shown to be effective in HER2-positive MBC, its effectiveness in *ERBB2*-mutated NSCLC remains unclear and is under clinical evaluation (NCT04579380).

**Table 2 biomolecules-15-01443-t002:** Efficacy of HER2-TKIs in patients with previously treated HER2-positive metastatic breast cancer.

Agents	Study	Phase	No. Patients with HER2-Positive MBC	ORR (%)	mPFS (Months)	mOS (Months)
Lapatinib (1250 mg daily) + Capecitabine (2000 mg/m^2^, 1–14 d/21 d)	Geyer et al. [50]	III	163	22	8.4	NA
Capecitabine (2500 mg/m^2^, 1–14 d/21 d)			161	14	4.1	NA
Neratinib (240 mg daily) + Capecitabine (750 mg/m^2^ twice daily 1–14 d/21 d)	NALA [49]	III	307	32.8	8.8	24.0
Lapatinib (1250 mg daily) + Capecitabine (1000 mg/m^2^ twice daily 1–14 d/21 d)			314	26.7	6.6	22.2
Pyrotinib (400 mg daily) + Capecitabine (1000 mg/m^2^ twice daily 1–14 d/21 d)	PHOEBE [53]	III	134	67.2	12.5	NR
Lapatinib (1250 mg daily) + Capecitabine (1000 mg/m^2^ twice daily 1–14 d/21 d)			132	51.5	6.8	26.9
Tucatinib (300 mg twice daily) + trastuzumab (6 mg/kg q3w) + capecitabine (1000 mg/m^2^ twice daily 1–14 d/21 d)	HER2CLIMB [56]	III	410	40.6	7.8	21.9
Placebo + trastuzumab (6 mg/kg q3w) + capecitabine (1000 mg/m^2^ twice daily 1–14 d/21 d)			202	22.8	5.6	17.4

HER2, human epidermal growth factor receptor; TKI, tyrosine kinase inhibitor; MBC, metastatic breast cancer; ORR, objective response rate; mPFS, median progression-free survival; mOS, median overall survival; NA, not available; 1–14 d/21 d, days 1–14 in every 21-day cycle.

**Table 3 biomolecules-15-01443-t003:** Efficacy of neratinib and pyrotinib in patients with *ERBB2*-mutated NSCLC.

Drug	Study	Phase	No. of Patients with *ERBB2* Mutation	ORR (%)	DCR (%)	mPFS (Months)	mOS (Months)	Grade 3/4 TRAE (%)
Neratinib (240 mg daily)	SUMMIT [47]	II	26 (previously treated)	3.8	42.3	5.5	NA	22% diarrhea
Neratinib (240 mg daily)	PUMA-NER-4201 [48]	II	17 (previously treated)	0	35	3.0	10.0	82% diarrhea
Neratinib (240 mg daily) + Temsirolimus (8 mg/weekly)			43 (previously treated)	19	51	4.1	15.8	86% diarrhea
Pyrotinib (400 mg daily)	Zhou et al. [52]	II	60 (previously treated)	30.0	85.0	6.9	14.4	28.3 20% diarrhea
	Song et al. [51]	II	78 (independent of prior treatments)	19.2	74.4	5.6	10.5	20.5 16.7% diarrhea
Pyrotinib (400 mg daily) + apatinib (250 mg daily)	PATHER2 [55]	II	33 (previously treated)	51.5	93.9	6.9	14.8	12.1 9.1% hypertension
Pyrotinib (320 mg daily) + inetetamab (6 mg/kg, q3w)	Huang et al. [58]	Ib	48 (independent of prior treatments)	25.0	84.1	5.5	NA	14.6

NSCLC, non-small cell lung cancer; ORR, objective response rate; DCR, disease control rate; mPFS, median progression-free survival; mOS, median overall survival, TRAE, treatment-related adverse events; NA, not available; q3w, every 3 weeks.

### 4.3. Poziotinib and Other Novel HER2-TKIs, Such as Zongertinib and BAY2927088

Zongertinib has received accelerated FDA-approval for previously treated patients with *ERBB2*-mutated advanced NSCLC in 2025 [59]. Poziotinib and other novel selective HER2 TKIs, such as BAY2927088, are not approved for clinical use in breast cancer and NSCLC in any country and are currently being evaluated for potential activity against *ERBB2*-mutated NSCLC.

It was demonstrated in a preclinical study that poziotinib, an irreversible EGFR/HER2 inhibitor, may tightly bind to the sterically hindered drug-binding pocket of HER2 A775_G776insYVMA and overcome the structural changes in HER2 induced by exon 20 insertions [60]. A multicenter, multicohort phase II study, ZENITH20, evaluated the efficacy of monotherapy with poziotinib in patients with *ERBB2*-mutated NSCLC. In this study, which included 90 previously treated patients with NSCLC with *HER2* mutations, poziotinib demonstrated an ORR, DCR, and mPFS of 27.8%, 70.0%, and 5.5 months, respectively [61]. Moreover, in treatment-naïve patients with *HER2* exon 20 insertions NSCLC, poziotinib treatment resulted in an ORR, DCR, and mPFS of 39%, 73%, and 5.6 months, respectively [62]. Thus, poziotinib has demonstrated moderate efficacy in *ERBB2*-mutated NSCLC. However, it was associated with considerable treatment-related adverse effects (TRAEs), especially those over grade 3 (Table 4). In the ZENITH20-2 study, 78.9% of patients who were treated with poziotinib had at least one grade 3 or higher TRAE, including rash (48.9%), diarrhea (25.6%), and stomatitis (24.4%). Furthermore, in the ZENITH20-4 study, at least one grade 3 or higher TRAE was observed in 71% of the patients. The most common grade 3 TRAEs were rash in 43%, stomatitis in 19%, and diarrhea in 18% patients. Grade 5 pneumonitis was reported in one patient. These TRAEs may represent EGFR-related toxic effects. In the NOV120101-203 study involving patients with HER2-positive MBC, the efficacy of poziotinib was favorable, but associated with a high incidence of grade 3 or higher EGFR-related toxicities (Table 5) [63]. More selective HER2 tyrosine kinase inhibitors are required for patients with HER2 alterations.

Novel selective HER2-TKIs, such as BAY2927088 and zongertinib were designed to have lower affinity for wild-type EGFR [64]. BAY2927088 is an oral, reversible HER2-TKI that showed a promising ORR of 70.5% in patients with *ERBB2*-mutated NSCLC who were naïve to HER2-targeted therapy and ORR of 35.3% in patients with *ERBB2*-mutated NSCLC who had received a HER2-targeted ADC [65]. Zongertinib is an oral, irreversible HER2-TKI that showed clinical benefit in previously treated *ERBB2*-mutated NSCLC with an ORR of 71%, DCR of 72%, and mPFS of 12.4 months in a cohort of HER2-targeted therapy naïve patients. Moreover, patients with *ERBB2*-mutated NSCLC who had received a HER2-targeted ADC demonstrated an ORR of 48% [66]. Based on this clinical trial, zongertinib received accelerated approval for previously treated patients with *ERBB2*-mutated advanced NSCLC [59]. Interestingly, the ORRs of patients who received BAY2927088 and zongertinib differed between HER2-targeted therapy-naïve patients and those who had received an HER2-targeted ADC. The ORR was relatively lower in patients who had previously received a HER2-targeted ADC than in those who were naïve to HER2-targeted therapy. This finding suggests the existence of cross-resistance between HER2 TKIs and HER2 ADCs. Consequently, the therapeutic sequence of HER2-TKI and HER2-ADC remains a critical issue, and it is unclear which agent should be used first to achieve a better outcome. As the HER2-TKI zongertinib exhibited a better safety profile, it may be preferable as first-line therapy. However, the effectiveness of TKI-ADC needs to be further investigated in NSCLC.

**Table 4 biomolecules-15-01443-t004:** Efficacy of selective HER2-TKIs poziotinib and zongertinib in patients with NSCLC.

Drug	Study	Phase	No. of Patients with *HER2* Mutation	ORR (%)	DCR (%)	mPFS (Months)	mOS (Months)	Grade 3/4 TRAE (%)	Key Grade 3/4 TRAE (%) (Rash/Diarrhea)
Poziotinib (16 mg daily)	ZENITH20-2 [61]	II	90 (previously treated)	27.8	70.0	5.5	NA	78.9	48.9/25.6
	ZENITH20-4 [62]	II	80 (Treatment naïve/HER2ex20ins)	39	73	5.6	NA	71	43.0/18.0
BAY2927088 (20 mg twice daily)	SOHO-01 [65]	I/II	44 (cohort D: HER2-targeted therapy naïve)	70.5	81.8	NA	NA	33.3	1.3/16.7
			34 (cohort E: previously treated with HER2-ADC)	35.3	52.9				
Zongertinib (120 mg or 240 mg daily)	Beamion Lung 1 [66]	Ib	75 (cohort 1: HER2-targeted therapy naïve)	71	72	12.4	NA	17 (8% increase ALT)	0/1.0
Zongertinib (120 mg daily)			31 (cohort 5: previously treated with HER2-ADC)	48	97	NA		3	0/0

HER2, human epidermal growth factor receptor; TKI, tyrosine kinase inhibitor; NSCLC, Non-small cell lung cancer; ORR, objective response rate; DCR, disease control rate; mPFS, median progression-free survival; mOS, median overall survival, TRAE, treatment-related adverse events; ADC, antibody-drug conjugate; NA, not available.

**Table 5 biomolecules-15-01443-t005:** Efficacy of HER2-TKIs in patients with HER2-positive metastatic breast cancer.

Drug	Study	Phase	No. Patients with HER2-Positive MBC	ORR (%)	DCR (%)	mPFS (Months)	mOS (Months)	Grade 3/4 TRAE (%)	Key Grade 3/4 TRAE (%) (Rash/Diarrhea)
Poziotinib (12 mg daily)	NOV120101-203 [63]	II	106 (previously treated)	25.5	73	4.04	NA	37.7	3.8/14.2

HER2, human epidermal growth factor receptor; TKI, tyrosine kinase inhibitor; ORR, objective response rate; DCR, disease control rate; mPFS, median progression-free survival; mOS, median overall survival, TRAE, treatment-related adverse events; NA, not available.

## 5. Clinical Use of Monoclonal Antibodies and ADCs Against HER2

### 5.1. The Anti-HER2 Monoclonal Antibodies Trastuzumab and Pertuzumab

Trastuzumab is a monoclonal antibody that binds to the HER2 extracellular domain and inhibits its dimerization, leading to the activation of downstream signaling. Trastuzumab monotherapy showed minimal efficacy in HER2-overexpressing NSCLC in the CALGB 39,810 study [67] (Table 6). Combination therapies with trastuzumab and platinum-based chemotherapy have been evaluated in patients with HER2-positive (mainly, HER2-overexpressing) NSCLC. Clinical trials of trastuzumab plus cisplatin/gemcitabine [68], carboplatin/paclitaxel [69] and docetaxel [70] showed disappointing clinical efficacy (Table 6). In a direct comparison, there was no additional benefit in adding trastuzumab to gemcitabine/cisplatin in patients with HER2-positive NSCLC [68]. The OS was similar to that of patients using carboplatin/paclitaxel alone as the first-line therapy [69] or docetaxel alone as the second-line therapy [70]. Therefore, trastuzumab at the same dose and schedule as that used in MBC cannot be recommended for NSCLC. Nowadays, for patients with HER2-positive (HER2-overexpressing) MBC, pertuzumab–trastuzumab–docetaxel or paclitaxel is recognized as the first-line treatment (Table 7) [71,72,73,74]. The Clinical Evaluation of Pertuzumab and Trastuzumab (CLEOPATRA) study showed that combining the anti-HER2 monoclonal antibodies trastuzumab and pertuzumab with docetaxel significantly prolonged OS in HER2-positive MBC, thus establishing the regimen as a first-line therapy [71,74]. The efficacy of the triple therapy, pertuzumab–trastuzumab–docetaxel, was evaluated in *ERBB2*-mutated NSCLC in the IFCT-1703 R2D2 study [75]. In this phase II study, 45 patients who were previously treated with platinum-based chemotherapy were treated with this triple therapy until disease progression was observed. An ORR of 29%, median PFS of 6.8 months, and median OS of 17.6 months were reported, demonstrating modest clinical efficacy.

### 5.2. The ADCs Trastuzumab Emtansine (T-DM1) and Trastuzumab Deruxtecan (T-DXd)

T-DM1 and T-DXd have been tested as anti-HER2 ADCs in patients with *ERBB2*-mutated or overexpressing NSCLC.

T-DM1 is an ADC that is loaded with the cytotoxic microtubule inhibitor DM1 via a non-cleavable linker. In 49 patients with HER2-overexpressing NSCLC, the efficacy of T-DM1 was limited, with an ORR of 20% and mPFS of 2.6 months [76] (Table 8). T-DM1 has been extensively evaluated in patients with *ERBB2*-mutated NSCLC, mainly in those harboring HER2 exon 20 insertions. However, it has demonstrated limited efficacy. T-DM1 yielded ORRs between 38.1 and 44%, with PFS varying between 2.8 and 5.0 months [77,78] (Table 8). In the phase III EMILIA study on patients with HER2-positive MBC who were previously treated with trastuzumab and taxane, T-DM1 achieved an ORR of 43.6%, PFS of 9.4 months, and OS of 30.9 months [79] (Table 9). Taken together, the ORR of 44% reported by Li et al. in *ERBB2*-mutated NSCLC is consistent with the rates reported by studies on breast cancer; however, the median PFS of 5 months in *ERBB2*-mutated NSCLC is shorter than the 10-month PFS in breast cancer.

T-DXd was recently approved by the FDA for patients with *ERBB2*-mutated NSCLC on the basis of promising clinical studies. T-DXd is an anti-HER2 ADC composed of trastuzumab loaded with the topoisomerase I inhibitor deruxtecan via a tetrapeptide-cleavable linker. In the DESTINY-Lung01 trial, T-DXd achieved an ORR of 55%, PFS of 8.2 months, and median OS of 17.8 months in 91 patients with *ERBB2*-mutated NSCLC [80] (Table 8), highlighting a major therapeutic advancement. Regarding the safety profile, 49% of patients who received T-DXd 6.4 mg/kg had grade 3 or higher drug-related adverse events, including hematologic and gastrointestinal adverse events; 26% patents had T-DXd related interstitial lung disease, with four patients developing grade 3 or higher interstitial lung disease, including two who died. Therefore, interstitial lung disease (ILD) is an important adverse effect of T-DXd requiring careful monitoring and management. In addition, the DESTINY-Lung02 trial validated the efficacy and safety of T-DXd in *ERBB2*-mutated NSCLC, although the dosage appeared to influence the response and safety, including the occurrence of drug-related ILD. Patients receiving a lower dose of 5.4 mg/kg, which is the same as the recommended and approved dosage in HER2-positive breast cancer, had an ORR, median PFS, and median OS of 49.0%, 9.9 months, and 19.5 months, respectively, while those on 6.4 mg/kg showed an ORR and median PFS of 56.0% and 15.4 months, respectively [81] (Table 8). Thus, the incidence of T-DXd-induced ILD/pneumonitis was lower at a dose of 5.4 mg/kg (12.9%) than at a dose of 6.4 mg/kg (28.0%). Considering the similar efficacy and reduced toxicity with regard to ILD/pneumonitis, the approved dosage of T-DXd is 5.4 mg/kg, However, the occurrence of pneumonitis should still be carefully considered because one patient died of ILD in each group of 5.4 mg/kg (102 patients) and 6.4 mg/kg (50 patients) in the DESTINY-Lung02 trial.

In HER2-overexpressing NSCLC, T-DXd use was associated with an ORR of 26.5% and 34.1% and median PFS of 5.7 months and 6.7 months in the cohorts receiving 6.4 mg/kg and 5.4 mg/kg, respectively [82] (Table 8). Even when the enrolled patients were heavily pretreated, the efficacy was modest. Regarding risk monitoring for pneumonitis and ILD with T-DXd, the all-grade incidence was 2% and 10% at doses of 5.4 mg/kg and 6.4 mg/kg, respectively. Although the study had a small sample size, a 5.4 mg/kg dose should be favored.

In patients with HER2-positive MBC, T-DXd exhibited an ORR of 79.7%, median PFS of 28.8 months, median OS of 52.6 months, and significantly greater efficacy compared to T-DM1 in the DESTINY-Breast03 trial [83,84] (Table 9). The data on the efficacy of T-DXd in HER2-overexpressing NSCLC and MBC suggest that while T-DXd has anti-tumor activity in HER2-overexpressing NSCLC, its efficacy is limited compared to that in HER2-positive MBC. A study has shown that patients with HER2-overexpressing NSCLC may have co-occurring somatic mutations, such as *EGFR* or *KRAS* [85]. Furthermore, HER2 overexpression or amplification may confer resistance to EGFR-TKIs in patients with NSCLC with *EGFR*-activating mutations [86]. Taken together, HER2-overexpressing NSCLC may have a varied pathogenesis and may not be dependent solely on HER2 signaling for tumor survival. Therefore, to identify patients with HER2-dependent tumor growth, a novel definition of HER2 overexpression or amplification is required in NSCLC instead of the existing threshold of HER2 IHC > 2+ and FISH greater than 2-fold.

Currently, the clinical potency of T-DXd in HER2-overexpressing NSCLC is being evaluated in the first-line setting in combination with other immunotherapeutic agents. The DESTINY-Lung03 is a phase Ib study evaluating the safety of a combination of T-DXd with durvalumab (an anti–PD-L1 antibody) and platinum-based chemotherapy as first-line therapy in patients with advanced or metastatic *ERBB2*-mutated NSCLC [87]. In the phase Ib DS8201-A-U106 (NCT04042701) trial, T-DXd was investigated in combination with pembrolizumab in treatment naïve patients with HER2-expressing NSCLC [88]. The results of these clinical trials on T-DXd will provide further data on its potential as a treatment option for HER2-overexpressing NSCLC.

Given the successful clinical studies on the efficacy of T-DXd in a later-line setting, T-DXd is currently being investigated in the first-line setting versus the standard treatment of chemotherapy plus pembrolizumab in *ERBB2*-mutated NSCLC in the phase III, DESTINY-Lung04 trial [89].

## 6. Discussion

Abnormalities of HER2 have been recognized as oncogenic drivers in NSCLC; however, their effects are not well understood. Three types of alterations in HER2 have been defined, mutation, amplification, and overexpression, which are more complicated in NSCLC than in breast cancer. Recently, *ERBB2* mutations, especially in exon 19 and 20, have emerged as a clear target for HER2-targeting therapies such as ADCs and TKIs in lung cancer.

T-DXd is the first approved drug with promising efficacy in patients with NSCLC harboring *ERBB2* mutations (Table 8). Earlier no TKIs were approved for targeting HER2 in patients with NSCLC because of their unsatisfactory efficacy and severe side-effects. However, the novel HER2-TKI zongertinib was recently approved as a treatment option for previously treated patients with *ERBB2*-mutated advanced NSCLC because of its favorable efficacy and safety profile (Table 4). In contrast to TKIs, the toxicity profiles of ADCs are driven by the effects of the payload. The payload of T-DXd is deruxtecan, a topoisomerase I inhibitor. T-DXd may cause life threatening ILD/pneumonitis. The overall incidence of all-grade ILD/pneumonitis cases was 11.4% in 14 studies that evaluated 1193 patients with different types of advanced solid malignancies [90]. However, the incidence and severity of ILD/pneumonitis vary among different malignancies, and NSCLC is associated with the highest incidence of ILD/pneumonitis (24.8%). The risk factors and underlying pathophysiology of T-DXd-induced ILD/pneumonitis must be identified to prevent its occurrence and to develop safe management strategies. The higher incidence in NSCLC has been attributed to preexisting lung damage from smoking, prior lung surgery, or lung radiation therapy. ILD/pneumonitis is treated according to the ILD management guidelines [91], including dose interruption, treatment discontinuation, and corticosteroid administration. The implementation of these ILD management guidelines reduced the rate of fatal ILD/pneumonitis due to T-DXd use in breast cancer from 2.7% in DESTINY-Breast01 [92] to 0.8% in DESTINY-Breast04 [93]. Despite the high anti-tumor activity of T-DXd, optimizing the management of ILD/pneumonitis is critical for its application in *ERBB2*-mutated NSCLC. For the early detection of IDL/pneumonitis, education of the care-providers and patients is critical. Careful monitoring by a multidisciplinary team is recommended, and should include symptom check, SpO2 level, chest X-rays, computed tomography, and pulmonary function tests. Because late detection after the onset of ILD/pneumonitis can lead to poor prognosis [94], careful monitoring may lead to early detection of low grade of ILD/pneumonitis. After detection, administration of T-DXd should be withheld and steroid treatment started. Therefore, translational or clinical studies are essential for preventing, detecting, and treating T-DXd-induced ILD and pneumonitis.

Novel strategies are required to improve the effectiveness of anti-HER2 targeted therapies. Combination therapies involving different targets or modalities may have synergistic effects. Combinations of ADCs and immune checkpoint inhibitors (ICIs), such as T-DXd with pembrolizumab or T-DXd with durvalumab, are being evaluated in clinical studies. In the umbrella (HUDSON) (NCT03334617) trial, patients with NSCLC with HER2 overexpression (n = 23) or *ERBB2* mutations (n = 20) who were previously treated with ICIs were administered a combination of T-DXd and durvalumab [95]. The ORR was 26.1% and 35%, mPFS was 2.8 and 5.7 months, and OS was 9.5 and 10.6 months in patients with HER2-overexpressing and *ERBB2*-mutated NSCLC, respectively. The incidence of ILD/pneumonitis was 9.3% for all grades (grade > 3: 7%; none were fatal) in all patients. The U106 study (NCT04042701), a phase IB study, tested a combination of T-DXd and pembrolizumab in ICI-treatment-naïve HER2-overexpressing (n = 22) or *ERBB2*-mutated (n = 33) patients with NSCLC [88]. The ORR was 54.5% and 66.7% and mPFS was 15.1 months and 11.3 months in HER2-overexpressing and *ERBB2*-mutated NSCLC, respectively. Adjudicated drug-related (T-DXd and/or pembrolizumab) ILD/pneumonitis of all grades occurred in 11 patients and one patient died. Thus, ILD or pneumonitis remains a notable adverse effect.

In the near future, the first-line therapy for patients with *ERBB2*-mutated NSCLC may change based on the results of phase III studies on T-DXd, zongertinib, and BAY2927088 compared to standard chemotherapy plus ICI therapy. Moreover, zongertinib showed encouraging tolerability and efficacy in HER2-altered solid tumors, such as MBC and metastatic gastric, gastroesophageal junction, or esophageal adenocarcinoma. Zongertinib plus T-DXd or T-DM1 are being tested in phase Ib/II trials [96]. Such a dual HER2-targeting regimen may be extended to advanced *ERBB2*-mutated NSCLC.

Moreover, new therapeutic agents are being developed, including bispecific antibodies (bsAbs) and allosteric inhibitors targeting HER2. The bsAbs simultaneously target dual antigens to enhance the effectiveness of a single monoclinal antibody, thus overcoming tumor heterogeneity and development of resistance [97]. Although there are no approved HER2-targeted bsAbs for patients with NSCLC or breast cancer, there are some FDA approved bsAbs targeting HER2 in some solid tumors. Zanidatamab is an anti-HER2 biparatropic bsAb that targets two different HER2 extracellular domains (ECD) of ECD2 and ECD4 simultaneously, and has been approved for previously treated, unresectable or metastatic HER2-positive biliary tract cancer [98,99]. Zenocutuzumab, a bsAb targeting both HER2 and HER3 receptors, blocks the binding of neuregulin1 (NRG1, HER3 ligand) to HER3 and inhibits HER2/HER3 heterodimerization. This agent is approved for the patients with advanced NSCLC and pancreatic adenocarcinoma harboring *NRG1* gene fusions [100]. Moreover, increasing attention is being paid to bsAb-drug conjugates as a novel therapeutic strategy with potentially greater effectiveness than the current ADCs. Crystallographic and mutagenesis studies have shown that HER2 is activated by *HER2* mutations occurring in allosteric sites outside the ATP binding site [101]. The allosteric inhibitor tuxobertinib (BDTX-189) a selective inhibitor of *EGFR* and *HER2* allosteric mutations is being investigated for the treatment of advanced solid tumors [102]. The clinical application of these inhibitors for *ERBB2*-mutated NSCLC may be challenging.

## 7. Conclusions

For the development of anti-HER2 therapies, *ERBB2* mutations are considered targetable oncogenic drivers. T-DXd has been approved for patients with pretreated *ERBB2*-mutated NSCLC. In fact, the efficacy of T-DXd in patients with pretreated NSCLC has encouraged a first-line phase III study in which T-DXd will be compared with chemotherapy plus ICI. In addition to T-DXd, zongertinib, a novel HER2-TKI, has been approved for previously treated patients with advanced *ERBB2*-mutated NSCLC by the FDA recently as it showed high efficacy and acceptable safety profiles. The novel HER2-TKIs may also be considered as first-line therapies for *ERBB2*-mutated NSCLC. Moreover, the development of novel therapeutic combinations for *ERBB2*-mutated NSCLC may be associated with improved treatment efficacy and tolerable safety profiles.

## Figures and Tables

**Figure 1 biomolecules-15-01443-f001:**
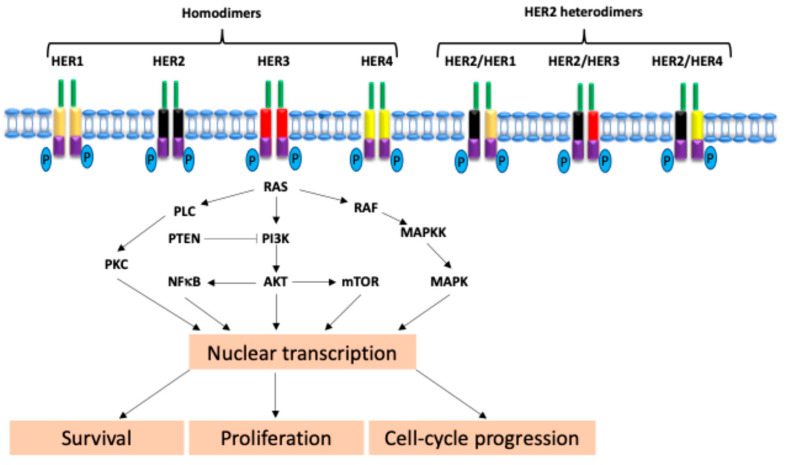
Schema of HER family signaling.

**Figure 2 biomolecules-15-01443-f002:**
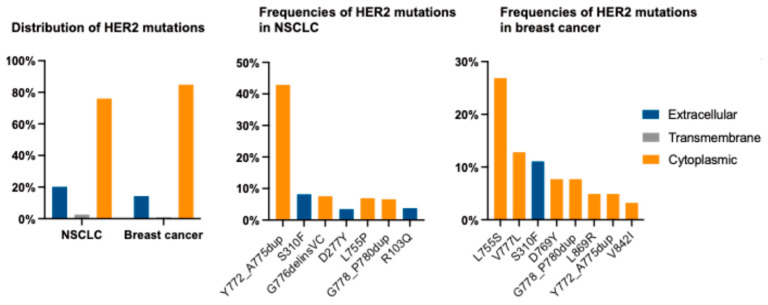
Distribution and frequencies of *ERBB2* mutations in patients with non-small cell lung cancer (NSCLC) and breast cancer.

**Figure 3 biomolecules-15-01443-f003:**
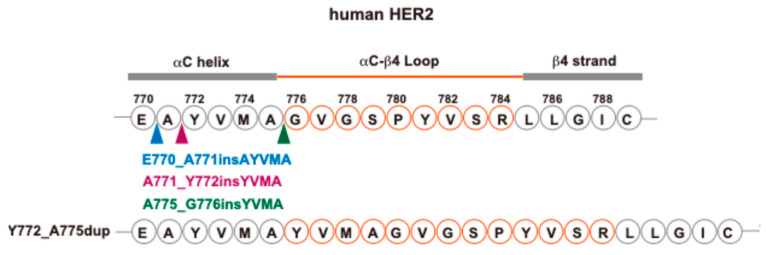
Sequence alignment of ERBB2 Y772_A775dup. Residues in the αC-β4 loop are shown in orange circles.

**Table 1 biomolecules-15-01443-t001:** Efficacy of the HER2-TKIs afatinib and dacomitinib in patients with *ERBB2*-mutated NSCLC.

Drug	Study	Phase	No. of Patients with *ERBB2* Mutation	ORR (%)	DCR (%)	mPFS (Months)	mOS (Months)	Grade 3/4 TRAE (%)
Afatinib (40 mg daily)	Fan et al. [41]	II	18 (EGFR/HER2 inhibitor naïve)	0	61.1	2.76	10.02	27.8
	Dziadziuszko et al. [42]	II	13 (previously treated)	7.7	53.8	3.7	12.9	NA
Afatinib (50 mg daily)	De Grève et al. [43]	II	7 (independent of prior treatments)	0	71	3.9	NA	NA
Dacomitinib (30–45 mg daily)	Kris et al. [44]	II	26 (any prior systemic therapy)	12.0	NA	3.0	9.0	53.3

HER2, human epidermal growth factor receptor; TKI, tyrosine kinase inhibitor; NSCLC, non-small cell lung cancer; ORR, objective response rate; DCR, disease control rate; mPFS, median progression-free survival; mOS, median overall survival, TRAE, treatment-related adverse events; NA, not available.

**Table 6 biomolecules-15-01443-t006:** Efficacy of anti-HER2 monoclonal antibodies in patients with NSCLC.

Agents	Study	Phase	Patient Population	N	ORR (%)	PFS (Months)	OS (Months)
Trastuzumab (first week: 4 mg/kg, 2 mg/kg weekly)	CALGB 39,810 [67]	II	HER2-positive (previously treated)	24	5.0	2.6	5.3
Cisplatin (75 mg/m^2^ day 1)/gemcitabine (1250 mg/m^2^ day 1, 8, q3w)	Gatzemeier et al. [68]	II	HER2-positive (first line)	50	41	7.0	NR
cisplatin (75 mg/m^2^ day 1)/gemcitabine (1250 mg/m^2^ day 1, 8) + Trastuzumab (first week: 4 mg/kg, 2 mg/kg weekly) q3w				51	36	6.1	12.2
Carboplatin (AUC 6, q3w)/paclitaxel (225 mg/m^2^ q3w) + Trastuzumab (first week: 4 mg/kg, 2 mg/kg weekly)	ECOG 2598 [69]	II	HER2-positive (first line)	53	24.5	3.25	10.1
Docetaxel (30 mg/m^2^ weekly for 6 weeks q8w) + Trastuzumab (first week: 4 mg/kg, 2 mg/kg weekly)	Lara et al. [70]	II	HER2-positive (second line)	13	8	4.3	5.7
Pertuzumab (1st cycle: 840 mg q3w, 420 mg q3w)/Trastuzumab (1st cycle 8 mg/kg q3w, 6 mg/kg q3w)/Docetaxel (75 mg/m^2^ q3w)	IFCT-1703 R2D2 [75]	II	HER2 mutation (previously treated)	45	29	6.8	17.6

HER2, human epidermal growth factor receptor; NSCLC, Non-small-cell lung cancer; ORR, objective response rate; PFS, progression-free survival; OS, overall survival; NR, not reached; q3w, every 3 weeks.

**Table 7 biomolecules-15-01443-t007:** Efficacy of anti-HER2 monoclonal antibodies in patients with MBC.

Agents	Study	Phase	Patient Population	N	ORR (%)	PFS (Months)	OS (Months)
Pertuzumab (1st cycle: 840 mg q3w, 420 mg q3w)/Trastuzumab (1st cycle 8 mg/kg q3w, 6 mg/kg q3w)/Docetaxel (75 mg/m^2^ q3w)	CLEOPATRA [71,74]	III	HER2-positive MBC (first-line therapy)	402	80.2	18.5	56.5
Placebo/Trastuzumab (1st cycle 8 mg/kg q3w, 6 mg/kg q3w)/Docetaxel (75 mg/m^2^ q3w)				406	69.3	12.4	40.8
Pertuzumab (1st cycle: 840 mg q3w, 420 mg q3w)/Trastuzumab (1st cycle 8 mg/kg q3w, 6 mg/kg q3w)/Docetaxel (the dose: investigator’s discretion, q3w)	PERUSE [72,73]	III	HER2-positive MBC (first-line therapy)	775	79	19.4	66.5
Pertuzumab (1st cycle: 840 mg q3w, 420 mg q3w)/Trastuzumab (1st cycle 8 mg/kg q3w, 6 mg/kg q3w)/Paclitaxel (dose at the investigator’s discretion, q3w)				588	83	23.2	64.0
Pertuzumab (1st cycle: 840 mg q3w, 420 mg q3w)/Trastuzumab (1st cycle 8 mg/kg q3w, 6 mg/kg q3w)/nab-Paclitaxel (dose at the investigator’s discretion, q3w)				65	77	19.2	70.9

HER2, human epidermal growth factor receptor; MBC, metastatic breast cancer; ORR, objective response rate; PFS, progression-free survival; OS, overall survival.

**Table 8 biomolecules-15-01443-t008:** Efficacy of HER2-ADC monotherapy in patients with NSCLC.

Agents	Study	Phase	Patient Population	N	ORR (%)	mPFS (Months)	mOS (Months)	Grade 3/4 TRAE (%)	Key TRAE Grade 1-2/3-5 (%) (ILD/Pneumonitis)
T-DM1 (3.6 mg/kg q3w)	Peters et al. [76]	II	*HER2* overexpression (previously treated)	49	20	2.6	12.2	22.4	0
	Iwama et al. [78]	II	*HER2* mutations (previously treated)	22	38.1	2.8	8.1	22.7	0
	Li et al. [77]	II	*HER2* mutations (treatment naïve and previously treated)	18	44	5.0	NA	6	0
T-DXd (6.4 mg/kg q3w)	DESTINY-Lung01 [80]	II	*HER2* mutations (previously treated)	91	55	8.2	17.8	49%	15.4/5.5 (one patient with Grade 5)
T-DXd (5.4 mg/kg q3w)	DESTINY-Lung02 [81]	II	*HER2* mutations (previously treated)	102	50	9.9	19.5	38.6	26/2 (one patient with Grade 5)
T-DXd (6.4 mg/kg q3w)				50	28	15.4	NE	58.0	10.9/2 (one patient with Grade 5)
T-DXd (6.4 mg/kg q3w)	DESTINY-Lung01 [82]	II	*HER2* overexpression (previously treated)	49	26.5	5.7	12.4	53	10/6 (one patient with Grade 5)
T-DXd (5.4 mg/kg q3w)				41	34.1	6.7	11.2	22	5/0

HER2, human epidermal growth factor receptor; NSCLC, non-small cell lung cancer; ORR, objective response rate; mPFS, median progression-free survival; mOS, median overall survival, TRAE, treatment-related adverse events; ADC, antibody-drug conjugate; NA, not available.

**Table 9 biomolecules-15-01443-t009:** Efficacy of HER2-ADC monotherapy in patients with previously treated advanced breast cancer.

Agents	Study	Phase	Patient Population	N	ORR (%)	PFS (Months)	OS (Months)	Grade 3/4 TRAE (%)	Key TRAE Grade 1-2/3-5 (%) (ILD/Pneumonitis)
T-DM1 (3.6 mg/kg q3w)	EMILIA [79]	III	HER2 positive MBC	495	43.6	9.4	30.9	40.8	0/0
Lapatinib (1250 mg daily/capecitabine (1000 mg/m^2^ twice daily 1–14 d/21 d)				496	30.8	5.8	25.1	57.0	0/0
T-DXd (5.4 mg/kg q3w)	DESTINY-Breast03 [83,84]	III	HER2 positive MBC	261	79.9	29.0	52.6	58.0	9.7/0.8
T-DM1 (3.6 mg/kg q3w)				264	36.9	7.2	42.7	52.1	1.9/0

HER2, human epidermal growth factor receptor; ORR, objective response rate; PFS, progression-free survival; OS, overall survival, TRAE, treatment-related adverse events; ADC, antibody-drug conjugate; MBC, metastatic breast cancer.

## Data Availability

Not applicable.

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
