# Peer review of "Recent Advances in the Development and Clinical Use of HER2 Inhibitors in Non-Small Cell Lung Cancer"

_biomolecules, 2025, doi:10.3390/biom15101443_

Round 1

Reviewer 1 Report

Comments and Suggestions for Authors

Summary & strengths
This is a clear and timely narrative review of HER2 alterations in NSCLC and the clinical development of HER2-targeted agents. The manuscript is well structured, clinically oriented, and up-to-date (e.g., T-DXd and recent HER2-TKIs, with sensible cross-talk to the breast-cancer experience). Tables are useful and concise. The piece will interest thoracic oncologists and translational researchers.

Comments/suggestions:

  1. State the review type and sources upfront.
    In the Abstract/Introduction (first paragraph), explicitly call this a narrative (not systematic) review and add a one-line note on information sources and time window searched. This will set reader expectations.

  2. Early definitions, then consistent terminology.
    In Section 1–2, provide a short boxed note defining (i) HER2 mutation, (ii) ERBB2 amplification, and (iii) HER2 overexpression, and emphasize that NSCLC lacks a universally accepted IHC/FISH standard. After that, use HER2 for protein and ERBB2 for the gene consistently, and unify abbreviations across text/tables.

  3. Tables – clarity/standardization.

    • Add a small footnote to each efficacy table specifying line of therapy, dose, and whether the population was HER2-mutated vs HER2-overexpressing.

    • Harmonize decimal places and “NA/NR” usage; ensure all abbreviations are defined in the footnotes.

    • Where space allows, a “Key grade ≥3 AEs” column (especially ILD/pneumonitis for T-DXd; rash/diarrhea for TKIs) would be very helpful.

  4. Safety and clinical guidance.
    Expand the safety paragraph on T-DXd-related ILD/pneumonitis to include a practical one-sentence monitoring tip (baseline CT, symptom check, early steroids, dose-hold/stop by grade). This increases utility without becoming a guideline.

  5. Sequencing and cross-resistance.
    You nicely note the different activity of TKIs in ADC-naïve vs ADC-exposed patients. Add a compact sentence on how this observation may influence sequencing (e.g., considerations when choosing between T-DXd and a HER2-TKI in earlier vs later lines).

  6. Figures.
    Consider a synthesizing figure/algorithm: “Assessment and treatment pathway for HER2-altered NSCLC” (testing → categorize as mutation vs amplification/overexpression → preferred therapies → key toxicities/notes). This would greatly enhance readability for clinicians.

  7. Minor style points.
    Remove a few redundancies across Intro/Discussion, check number consistency between text and tables, and perform a light copy-edit for hyphenation (e.g., “HER2-mutated”), en-dash ranges, and consistent use of percent signs.

Overall, this is a solid and clinically useful review; the above are polishing suggestions to strengthen clarity and practical impact.

Author Response

Thank you for reviewing our manuscript biomolecules-3900679, entitled “Recent advances in the development and clinical use of HER2 inhibitors in non-small cell lung cancer.” We are very grateful to the Editors and reviewers for their positive feedback and helpful comments for improving our manuscript. The point-by-point response to the two reviewers’ comments is given below. All changes in the manuscript have been highlighted in yellow.

Response to Reviewer 1’s comments:
We are pleased that the reviewer felt that “This is a clear and timely narrative review of HER2 alterations ….” and that “The piece will interest thoracic oncologists and translational researchers.”

Comments/suggestions: 

1.    State the review type and sources up front. 
In the Abstract/Introduction (first paragraph), explicitly call this a narrative (not systematic) review and add a one-line note on information sources and time window searched. This will set reader expectations. 
Response:
    We agree with the reviewer on the need to provide a statement regarding the review type and information sources in the abstract and introduction. Accordingly, we have made some changes in the abstract (lines 20-21) and introduction (lines 46-47).

2.    Early definitions, then consistent terminology.
In Section 1–2, provide a short-boxed note defining (i) HER2 mutation, (ii) ERBB2 amplification, and (iii) HER2 overexpression, and emphasize that NSCLC lacks a universally accepted IHC/FISH standard. After that, use HER2 for protein and ERBB2 for the gene consistently, and unify abbreviations across text/tables.
Response:
    We apologize for the confusion caused by the inconsistent expression of ERBB2 and HER2. In accordance with your suggestion, we have made edits to indicate that ERBB2 indicates the gene and HER2 the protein throughout the manuscript. We have also provided information regarding the lack of standard FISH/IHC guidelines in NSCLC for measuring ERBB2 gene amplification and overexpression, respectively, in Section 3.2 (lines 154–156).

3. Tables – clarity/standardization.
Add a small footnote to each efficacy table specifying line of therapy, dose, and whether the population was HER2-mutated vs HER2-overexpressing.
Harmonize decimal places and “NA/NR” usage; ensure all abbreviations are defined in the footnotes.
Where space allows, a “Key grade ≥3 AEs” column (especially ILD/pneumonitis for T-DXd; rash/diarrhea for TKIs) would be very helpful.
Response:
    We appreciate the reviewer’s thoughtful comment on the tables. We have added the patients’ therapeutic background and the population type (whether HER2-mutated or HER2-overexpressing). Furthermore, “NA” and “NR” have been defined as “not available” and “not reached,” respectively, in the footnotes of the tables. Incorporation of “Key grade >3 AEs” is a wonderful suggestion, and this has been included in Tables 4, 5, 8, and 9.

4.    Safety and clinical guidance.
Expand the safety paragraph on T-DXd-related ILD/pneumonitis to include a practical one-sentence monitoring tip (baseline CT, symptom check, early steroids, dose-hold/stop by grade). This increases utility without becoming a guideline.
Response:
    We thank the reviewer for the valuable comment and agree with the suggestion. We have added information regarding the monitoring of T-DXd-related ILD/pneumonitis in the Discussion (lines 462–568).

5. Sequencing and cross-resistance.
You nicely note the different activity of TKIs in ADC-naïve vs ADC-exposed patients. Add a compact sentence on how this observation may influence sequencing (e.g., considerations when choosing between T-DXd and a HER2-TKI in earlier vs later lines).
Response:
    We appreciate the reviewer’s thoughtful comments. According to the results of the Beamion Lung 1 trial on zongertinib, HER2-TKI should be considered an earlier line therapy than T-DXd for patients with HER2-mutated NSCLC because its toxicity is not as serious as the ILD/pneumonitis linked to T-DXd. It is very important to continue the therapy safely in patients with advanced NSCLC. The main goals of treating patients with advanced NSCLC are prolonging the survival and relieving symptoms. However, depending on the resistance mechanism(s) to zongertinib and T-DXd, the optimal sequence of the anti-HER2 therapies of HER2-ADC and HER2 -TKI needs to be clarified. We have added this information in Section 4.3 (lines 306–310). 

6. Figures.
Consider a synthesizing figure/algorithm:
“Assessment and treatment pathway for HER2-altered NSCLC”
(testing →categorize as mutation vs amplification/overexpression →preferred therapies → key toxicities/notes). This would greatly enhance readability for clinicians.
Response:
    We appreciate the reviewer’s thoughtful comment and agree that it is important to improve the readability of the manuscript for readers. We have tried to address the arrangement of the relevant sections in accordance with your comment; however, we have included the pivotal clinical studies in breast cancer to obtain insights for identifying more effective therapies for NSCLC. Therefore, it would be difficult to rearrange these paragraphs in the manuscript.

7. Minor style points.
Remove a few redundancies across Intro/Discussion, check number consistency between text and tables, and perform a light copy-edit for hyphenation (e.g., “HER2-mutated”), en-dash ranges, and consistent use of percent signs.
Response:
    We apologize that there are some redundant expressions in a few sections and some editorial inconsistencies. These errors have been corrected in the tables and text.

Reviewer 2 Report

Comments and Suggestions for Authors

The review titled “Recent advances in the development and clinical use of HER2  inhibitors in non-small cell lung cancer” summarizes and discusses that the HER2 gene alterations are recognized drivers and therapeutic targets in non-small cell lung cancer, with recent FDA approval of HER2-specific therapies like zongertinib showing promising results. While HER2-targeted treatments are more advanced in breast cancer, insights from these studies may guide more effective therapies for HER2-altered NSCLC. The manuscript is well-organized and informative; however, some details should be clarified:

  1. It is suggested that using a table comparing HER2-targeted therapies, for example, TKIs, ADCs and antibodies, in NSCLC and breast cancer with columns for agent, mechanism, approved indication, and clinical outcomes would greatly enhance the clarity and impact.
  2. The biology is well explained but consider briefly comparing the signaling implications of HER2 alterations in NSCLC vs. breast cancer to maintain relevance to the review’s central theme.
  3. Lines 110–130, the explanation is good, but the transition from NSCLC to breast cancer can be more clearly marked. Also, briefly summarize the clinical implications of HER2 mutations in breast cancer, given they are not yet standard targets.
  4. In section 4.2, it’s suggested to reinforce that neratinib is active in HER2-amplified breast cancer but ineffective in HER2-mutant NSCLC—highlighting the need for mutation-specific targeting.
  5. Consider a short note about how HER2 signaling is context-dependent—for example, amplified HER2 drives breast cancer, but HER2 mutations activate kinase signaling in NSCLC with different downstream effects.
  6. Please discuss ongoing development of allosteric or bispecific HER2 TKIs, or agents designed to overcome resistance mutations.

Author Response

Thank you for reviewing our manuscript biomolecules-3900679, entitled “Recent advances in the development and clinical use of HER2 inhibitors in non-small cell lung cancer.” We are very grateful to the Editors and reviewers for their positive feedback and helpful comments for improving our manuscript. The point-by-point response to the two reviewers’ comments is given below. All changes in the manuscript have been highlighted in yellow.

Response to Reviewer 2’s comments:
We are pleased that the reviewer felt that “The manuscript is well-organized and informative.”

Comments/suggestions:

1.    It is suggested that using a table comparing HER2-targeted therapies, for example, TKIs, ADCs and antibodies, in NSCLC and breast cancer with columns for agent, mechanism, approved indication, and clinical outcomes would greatly enhance the clarity and impact.
Response:
We appreciate the reviewer’s thoughtful comment. We agree with your suggestion that tables in this manuscript should display a comparison of HER2-targeted therapies, such as TKIs, ADCs, and antibodies in NSCLC and breast cancer with separate columns for agents, clinical study names, patients with HER2 alterations, and clinical outcomes. In accordance with this suggestion, we have included the dosage of the inhibitors in the tables.

2.    The biology is well explained but consider briefly comparing the signaling implications of HER2 alterations in NSCLC vs. breast cancer to maintain relevance to the review’s central theme.
Response:
    We agree with the suggestion of adding a clarification regarding the signaling implications of HER2 alterations in NSCLC vs. breast cancer. We have addressed this issue by adding an explanation regarding the differences in HER2 activation in NSCLC and breast cancer in Section 2 (lines 80–85).

3.    Lines 110–130, the explanation is good, but the transition from NSCLC to breast cancer can be more clearly marked. Also, briefly summarize the clinical implications of HER2 mutations in breast cancer, given they are not yet standard targets.
Response:
    We agree the reviewer’s comment that “the transition from NSCLC to breast cancer can be more clearly marked.” Accordingly, we have made some changes and explained the clinical implications of HER2 mutation in breast cancer in Section 3.1 (lines 118–119, 132–134).

4.    In section 4.2, it’s suggested to reinforce that neratinib is active in HER2-amplified breast cancer but ineffective in HER2-mutant NSCLC—highlighting the need for mutation-specific targeting.
Response:
We thank the reviewer for the expert comment and agree with the reviewer. We have added a sentence to highlight the need for mutation-specific targeting in Section 4.2 (lines 228–229).

5.    Consider a short note about how HER2 signaling is context-dependent—for example, amplified HER2 drives breast cancer, but HER2 mutations activate kinase signaling in NSCLC with different downstream effects.
Response:
    We thank the reviewer the insightful comment. We have added an explanation regarding the differences in HER2 activation between NSCLC and breast cancer in Section 2 (lines 80–86).

6.    Please discuss ongoing development of allosteric or bispecific HER2 TKIs, or agents designed to overcome resistance mutations.
Response:
    We thank the reviewer for this valuable suggestion. Accordingly, information on allosteric inhibitors or bispecific antibodies has been added in the Discussion (lines 496–514).

Round 2

Reviewer 2 Report

Comments and Suggestions for Authors

All my comments were addressed appropriately.